

# Comparing and characterizing scapular muscle activation ratios in males and females during execution of common functional movements

Angelica E. Lang[1,2], Annaka Chorneyko[3] and Vivian Heinrichs[3]

[1] Canadian Centre for Rural and Agricultural Health, University of Saskatchewan, Saskatoon, Saskatchewan, Canada
[2] Department of Medicine, University of Saskatchewan, Saskatoon, SK, Canada
[3] College of Medicine, University of Saskatchewan, Saskatoon, Saskatchewan, Canada

Corresponding author
Angelica E. Lang,
angelica.lang@usask.ca

## ABSTRACT

**Background**. The shoulder complex relies on scapular movement controlled by periscapular muscles for optimal arm function. However, minimal research has explored scapular muscle activation ratios during functional tasks, nor how they might be influenced by biological sex. This investigation aims to characterize how sex impacts scapular muscle activation ratios during functional tasks.

**Methods**. Twenty participants (ten females, ten males) were assessed with surface electromyography (EMG) and motion tracking during seven functional tasks. Activation ratios were calculated from normalized EMG for the three trapezius muscles and serratus anterior. Scapular angles were calculated using a YXZ Euler sequence. Two-way mixed methods ANOVAs ($p < .05$) were used to assess the effects of sex and humeral elevation level on ratios and angles.

**Results**. Sex-based differences were present in the Tie Apron task, with males exhibiting higher upper trapezius/lower trapezius and upper trapezius/middle trapezius ratios than females. Males also demonstrated decreased internal rotation in this task. Other tasks showcased significant sex-based differences in scapular upward rotation but not in activation ratios. Humeral elevation generally demonstrated an inverse relationship with scapular muscle activation ratios.

**Conclusions**. This study highlights sex-based differences in scapular muscle activation ratios during specific functional tasks, emphasizing the need to consider sex in analyses of shoulder movements. Normative activation ratios for functional tasks were provided, offering a foundation for future comparisons with non-normative groups. Further research is warranted to confirm and explore additional influencing factors, advancing our understanding of shoulder activation and movement in diverse populations.

## INTRODUCTION

The shoulder complex is a unique articulation consisting of four different joints allowing for a wide range of available motion. Movement at the scapulothoracic joint represents an important aspect of healthy shoulder motion. The scapula is an anchor for the arm, and

scapular motion is required to position the glenohumeral joint to optimize arm and hand positioning and function. The periscapular muscles are crucial to this system (*Cricchio & Frazer, 2011*). Alterations to the activation of these scapular muscles, and the subsequent kinematics, are associated with shoulder musculoskeletal disorders (*Keshavarz et al., 2017*; *Ludewig & Cook, 2000*; *Ludewig & Reynolds, 2009*; *Phadke, Camargo & Ludewig, 2009*; *Seitz et al., 2011*) encouraging the need for continued improved understanding of scapular muscle activation and movement.

Relative activation of scapular muscles is important to understanding shoulder biomechanics. Force couples, in which two muscles in opposing directions work together to create movement, are necessary for healthy scapular biomechanics during arm movement (*Contemori, Panichi & Biscarini, 2019*; *Thigpen et al., 2010*). Specifically, the force couples of the upper trapezius and lower trapezius and upper trapezius and serratus anterior work to create scapular upward rotation and posterior tilt which are necessary for typical, healthy arm movement. Activation ratios are frequently used to describe these force couples and may give insight into function or injury risk (*Cordasco et al., 2010*; *Moeller, Huxel Bliven & Snyder Valier, 2014*). However, most investigations of these scapular activation ratios have focused on training exercises or planar arm elevation (*Schory et al., 2016*; *Spall, Ribeiro & Sole, 2016*). Scapular activation and ratios are not well defined in functional tasks, despite their importance to work, daily life, and overall functioning.

Biological sex influences several aspects of anatomy and physiology. Notably, size and strength often differ by sex (*Kritzer et al., 2024*), but other anthropometric, muscle physiology, motor control, and muscle activation measures may also vary between men and women (*Côté, 2012*), necessitating further comparisons. Specifically, magnitude and coordination of shoulder muscle activations differ between the sexes in an assortment of movements and tasks for shoulder muscles (*Anders et al., 2004*; *Bouffard et al., 2019*; *Martinez et al., 2019*), with females often demonstrating higher muscular demands than males, which is possibly related to the higher rate of injury in females (*Wijnhoven et al., 2006*). The trapezius muscle is also influenced by sex: females may activate all trapezius sections more than males during arm elevation (*Szucs & Borstad, 2013*), while males may activate select sections higher after an office work task (*Szucs & Molnar, 2017*). More research is needed to determine how sex influences scapular muscle activation in a wider range of functional tasks.

Taken together, there is a gap in knowledge surrounding how the activation ratios of important scapular force couples may differ between sexes during functional tasks. Activation ratios are of particular interest to compare as they represent a measure of within-person relative activation patterns. Ratios may be less affected than individual muscle activations by known strength differences between sexes (*Murray et al., 1985*). In addition, normative activation ratios during functionally-relevant movements need to be defined for comparison in future research. These data will provide a valuable reference point for further comparisons by sex and to help interpret research into performance of different pathological groups. Therefore, the purpose of this study was to define and compare activation ratios of scapular muscles (upper trapezius/lower trapezius (UT/LT), upper trapezius/serratus anterior (U T/SA), upper trapezius/middle trapezius (UT/MT),

and middle trapezius/lower trapezius (MT/LT)) in healthy males and females during a functional task protocol. A secondary purpose was to compare scapular orientations to provide context to ratio data. The primary hypothesis was that women would have overall higher activation ratios, due to increased upper trapezius activity that may be connected to increased rates of injury.

## MATERIALS & METHODS

### Participants

Ten females and ten males were recruited from a convenience sample ([mean(SD)] age: 24(2) years, height:1.7(0.1) m, weight: 79.1(16.6) kg, 20 right-handed). An *a priori* between-factors repeated measures ANOVA sample size calculation using an effect size of 0.53 (*Waslen, Friesen & Lang, 2023*), power set to 0.8, alpha set to .05, 2 groups with 5 measurements (30° increments from 30° to maximum) estimated that a total sample size of 20 participants was required (*Faul et al., 2007*); observed power in these data was an average of 0.82. Exclusion criteria included (1) under the age of 18, (2) presence of upper body pain or musculoskeletal impairments, (3) previous shoulder surgery, (4) presence of other health-related disorders, (5) inability to raise arms overhead, and (6) allergies to adhesives. All participants provided informed, written consent and then completed a brief questionnaire to characterize their upper limb function (QuickDASH). The average QuickDASH score (out of 100) was 2.0 (range: 0–15.9), indicating this sample had little to no upper limb impairments.The study procedures were approved by the University of Saskatchewan's ethics board (Bio #3796).

### Instrumentation

Shoulder muscle activity and motion were measured for all participants. The dominant side (right for all participants) data are reported in this study. Surface electromyography (EMG) sensors (Delsys Trigno™ Wireless EMG sensors; Delsys, Inc, Natick, MA) were placed over the muscle bellies of the upper trapezius, middle trapezius, lower trapezius, and serratus anterior based on previously published standards (*Criswell, 2010*). Before placing the electrodes, the area was shaved and cleansed with isopropyl alcohol. Participants completed maximum voluntary contractions (MVCs) for normalization. One round of four MVCs was performed (Table 1) to minimize potential fatigue effects, as this study was part of a larger protocol. Each MVC was 5 seconds. Participants were instructed to ramp up to their maximum intensity in the first two seconds and maintain the exertion for the remaining time. At least one minute of rest was allowed between each MVC.

Individual and clustered reflective markers were placed on the torso, scapula, and humerus to track shoulder movement based on International Society of Biomechanics (ISB) standards (*Wu et al., 2005*). The scapula was tracked with an acromial marker cluster, and a double calibration method was applied (*Brochard, Lempereur & Rémy-Néris, 2011*; *Friesen et al., 2023*; *Lang, 2023*). Motion was tracked with a 10 camera optoelectronic system (Vicon Motion Systems, Oxford, UK) sampling at 100 Hz. EMG data were synced with motion capture data and sampled at 2,000 Hz.

**Table 1  Maximum voluntary contraction positions (*Garcia et al., 2023*; *Mackay et al., 2023*).**

| Muscle | Description |
| --- | --- |
| Upper trapezius | The participant was seated with their arm abducted to 90°, elbow bent to 90° and forearm parallel with the floor. Participants were directed to raise and shrug their shoulder with manual resistance applied just above the elbow. |
| Middle trapezius | The participant was seated with their arm abducted to 90°, elbow bent to 90° and forearm parallel with the floor. Participants were directed to push their arm backwards and retract their scapula with manual resistance applied just above the elbow. |
| Lower trapezius | The participant laid prone on a treatment table with their arm abducted to 120° with thumb up and elbow straight, parallel to the floor. They were directed to lift the arm upward with resistance applied at the wrist. |
| Serratus anterior | The participant stood, slightly bent forward with torso curled anteriorly and scapulae protracted, with hands clasped. They were directed to push their hands together and pull the elbows downward, providing their own resistance. |

**Table 2  Work related activities and functional tasks (WRAFT) protocol.**

| Task | Description |
| --- | --- |
| Comb hair | Hold a comb in their hand resting on their lap, bring it to the forehead, and pretend to comb from front to back. |
| Wash axilla | Hold a washcloth in their hand resting on their lap, bring it to the opposite anterior axilla, and pretend to wash. |
| Tie apron | Start standing with arms by their sides and holding a ribbon with both hands, bring hands to the waist level, reach behind the back until the hands meet, and pretend to tie the ribbon. |
| Overhead reach | While seated with hand resting on the shelf, lift a 1 kg object at table height to a target on a shelf 1.5 m off the ground. |
| Forward transfer | While seated with hand resting on table, transfer a 1 kg object at table height to a mark 50 cm forward. |
| Floor to waist lift | While standing with arms by their sides, lift a standard sized milk crate with an 8 kg load with both hands from the floor to a shelf at waist height. |
| Overhead lift | While standing with hands on the crate, lift a standard sized milk crate with an 8 kg load with both hands from a shelf at waist height to a shelf at forehead height. |

## Protocol

This study utilized the Work-Related Activities and Functional Task (WRAFT) protocol (Table 2), which is described in further detail by a previously published work (*Friesen et al., 2023*). Participants completed three repetitions of each task (three on each side for unilateral tasks), in a series of seven tasks that reproduce activities of daily living and working. Instructions and demonstrations were provided throughout the protocol.

## Analysis

All data were processed with custom MATLAB codes. Marker data were filtered with a low pass, fourth-order, zero-lag Butterworth filter (cutoff = 6 Hz) (*Murgia, Kyberd & Barnhill, 2010*; *Winter, 2009*), and local coordinate systems for the torso and humerus were defined based on ISB standards (*Wu et al., 2005*). The YXZ Euler sequence was used to calculate scapular angles (internal rotation, upward rotation, tilt). Humeral elevation was calculated as the angle between the long axes of the torso and humerus, as this method is effective for calculating consistent humeral elevation, unaffected by plane (*Friesen et al., 2023*). The timing (frame number) of 30° increments of humeral elevation (30°, 60°, 90°, 120°, maximum) were defined for scapular orientation and EMG analysis. As the functional tasks did not all have the same level of humeral movement, only elevation levels relevant to each task were evaluated. Specifically, for the Wash Axilla, Tie Apron, Forward Transfer, and Floor Lift, only 30°, 60°, and maximum were assessed, while only 60°, 90°, and maximum were assessed for the Overhead Reach and Overhead Lift (*Friesen et al., 2023*). Scapular orientations were extracted at these humeral elevation levels.

All EMG data were first filtered with a high pass Butterworth filter with a cutoff of 30 Hz (*Drake & Callaghan, 2006*) to reduce heart rate artifacts. Next, a second order, single pass Butterworth filter with a cutoff of 3 Hz linear enveloped the full wave rectified data (*Waite, Brookham & Dickerson, 2010*). The linear enveloped signal for each muscle was normalized to the linear enveloped maximum value from the MVCs. Activation ratios were calculated for each trial from the normalized data: UT/LT, UT/MT, UT/SA, and MT/LT. Values over 1 indicate higher relative upper trapezius activity or middle trapezius activity (MT/LT). Activation ratios were down sampled to match motion capture data, and values at each humeral elevation level (30°, 60°, 90°, 120°, maximum) were extracted for analysis.

Two-way mixed methods ANOVAs ($p < .05$) were used to assess the effects of biological sex and humeral elevation (30°, 60°, 90°, 120°, max, as applicable) on activation ratios (UT/LT, UT/SA, UT/MT, and MT/LT) and scapular kinematics (internal rotation, upward rotation, tilt).

## RESULTS

### Sex effects

Interaction and sex main effects were present for muscle activation ratios and scapular kinematics during the Tie Apron task. UT/LT (Fig. 1) and UT/MT (Fig. 2) ratios were significantly influenced by the interaction of humeral elevation and sex during this task ($p = .005 - .019$, Cohen's $f = .55–.67$) (Fig. 1), with the ratio increasing more in males than females as humeral movement increased. There was also a main effect of sex for the UT/SA (Fig. 3) ($p = .038$, $f = .33$). Finally, there was a sex main effect on scapular internal rotation in the same task, with males showing decreased internal rotation ($p = .036$, $f = .42$) (Table 3).

There were no other significant sex effects (main or interaction) for activation ratios (Figs. 1–4). However, in the Comb Hair, Overhead Reach, and Forward Transfer tasks, females demonstrated less overall scapular upward rotation throughout the entire

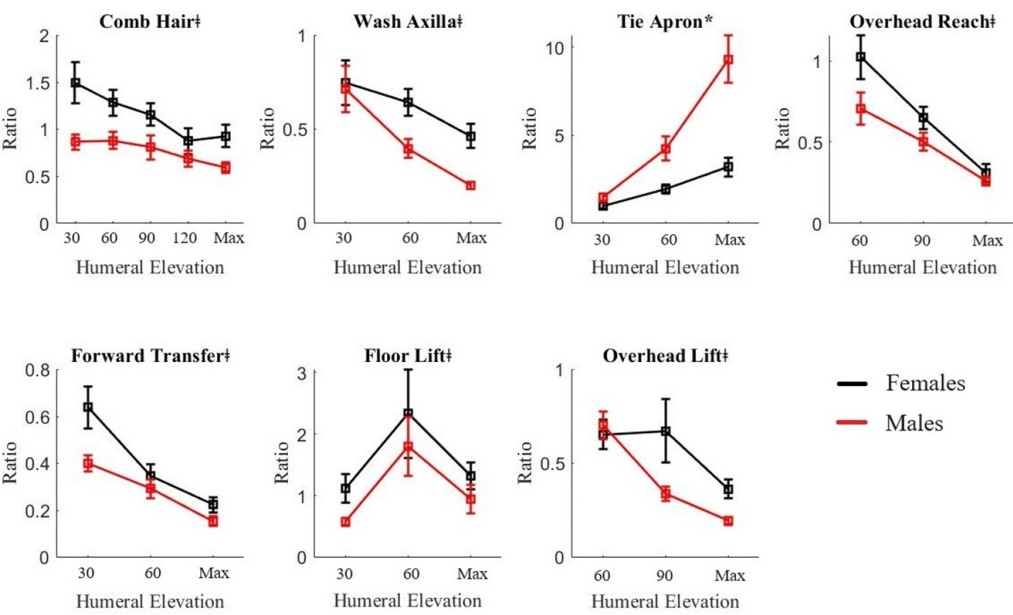

**Figure 1 Mean +/- 95% CI UT/LT activation ratio for females (black line) and males (red line) for all functional tasks.** An asterisk (*) indicates significant interaction of sex and humeral elevation level ($p <$ .05); # indicates significant main effect of sex ($p < .05$); ‡ indicates significant main effect of humeral elevation level ($p < .05$).

movement ($p = $ .010–.038, $f = $ .34–.49) (Table 3). While muscle activation results did not reach the significance threshold, the UT/LT ratio was visibly higher for females in these tasks (Fig. 1).

All mean activation ratios and individual muscle activations are included by sex and humeral elevation for each task in the Supplemental Material.

### Humeral level

Scapular muscle activation ratios varied with humeral elevation during all functional tasks (Figs. 1–4). In most tasks and ratios, there was an inverse relationship with humeral elevation. There were a few exceptions: notably, the Tie Apron task ratios generally increased with humeral elevation, indicating greater relative UT or MT activity. In the Comb Hair task, there were no significant differences with humeral elevation for the UT/SA, UT/MT, and MT/LT.

## DISCUSSION

Select shoulder muscles work together in couples to move the scapula for shoulder and hand positioning to complete desired tasks. These couples can be represented by activation ratios of the muscle activity measures, which may provide insight into movement patterns. This study compared scapular muscle activation ratios between sexes in a range of functional tasks in a healthy group. Some differences were found between females and males in the Tie Apron, but most tasks demonstrated activity not different between sexes. These

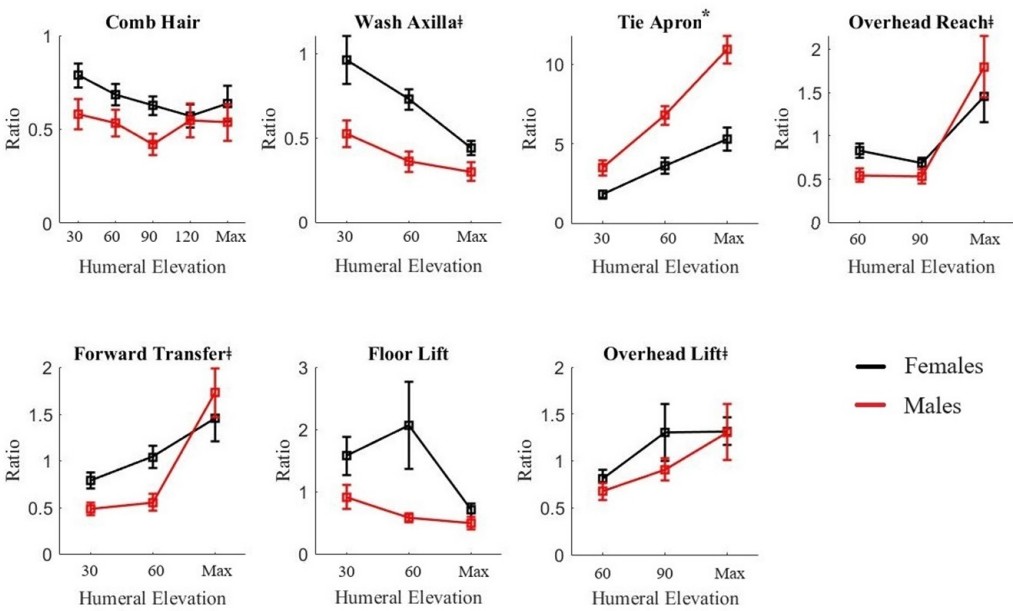

**Figure 2** **Mean +/- 95% CI UT/MT activation ratio for females (black line) and males (red line) for all functional tasks.** An asterisk (*) indicates significant interaction of sex and humeral elevation level ($p < .05$); # indicates significant main effect of sex ($p < .05$); ‡ indicates significant main effect of humeral elevation level ($p < .05$).

data also provide definitions of normative activation ratios for functional tasks for future comparisons.

There were sex-based differences present in activation ratios in only one task (Tie Apron), while scapular kinematics differed between sexes in four out of the seven tasks (Tie Apron, Comb Hair, Overhead Reach, and Forward Transfer). Males demonstrated higher ratios for the UT/LT and UT/MT than females in the Tie Apron task, indicating that males activate their upper trapezius more than females relative to the other muscles in the couples. These differences corresponded with greater scapular external rotation in males. The Tie Apron task is akin to perineal care or a hand to back pocket movement, which requires extension of the humerus (*van Andel et al., 2008*). The high activation of the upper trapezius in males could be a result of lower flexibility at the glenohumeral joint in males; females are known to have higher range of motion and joint laxity (*Barnes, Van Steyn & Fischer, 2001*; *Larsson, Baum & Mudholkar, 1987*; *Maier et al., 2022*) which could allow them to complete the movement at the glenohumeral joint, while males need to retract their shoulders. Additionally, males leveraged greater scapular upward rotation to complete the three unilateral tasks (Comb Hair and Overhead Reach and Froward Transfer), which was not explained by significant activation pattern differences. Visibly lower ratios were present for males in both tasks, though, which would be expected with higher upward rotation. It is possible that, while this study was powered to detect between-group kinematic differences, a larger sample size may be required to detect all significant EMG differences. Regardless, these scapular changes (more upward rotation, less internal rotation) are both considered

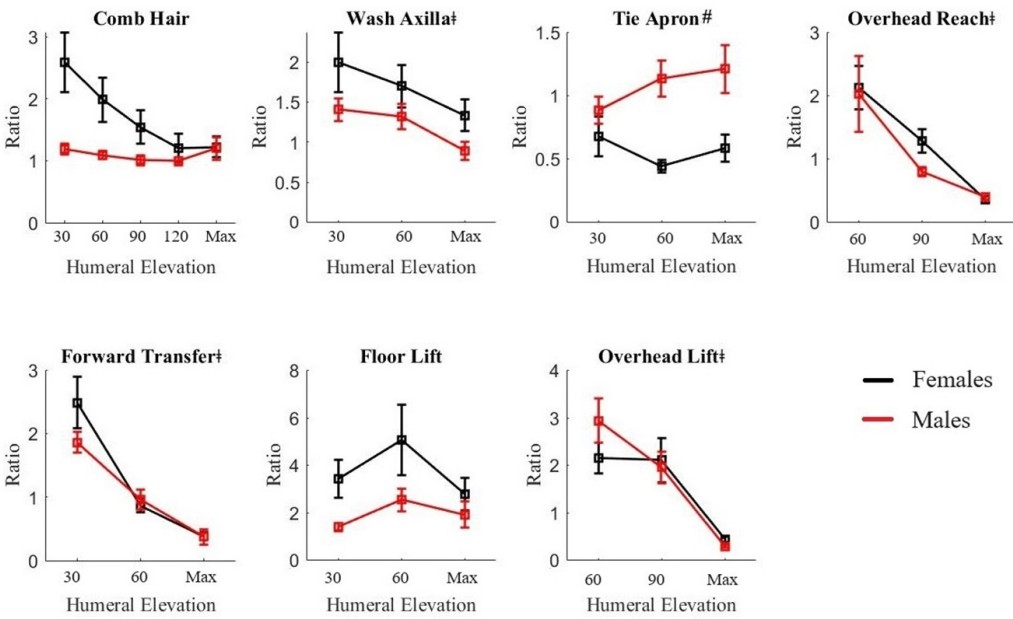

**Figure 3** **Mean +/- 95% CI UT/SA activation ratio for females (black line) and males (red line) for all functional tasks.** An asterisk (*) indicates significant interaction of sex and humeral elevation level ($p < .05$); # indicates significant main effect of sex ($p < .05$); ‡ indicates significant main effect of humeral elevation level ($p < .05$).

**Table 3** **Scapular angles (in degrees) [mean (95% confidence interval)] with significant differences between sexes ($p < .05$).**

| Angle | Task | | 30° | 60° | 90° | 120° | Maximum |
|---|---|---|---|---|---|---|---|
| Internal rotation | Tie apron | F | 39 (2) | 36 (2) | – | – | 35 (2) |
| | | M | 34 (1) | 24 (2) | – | – | 27 (2) |
| Upward rotation | Comb hair | F | 3 (1) | 13 (1) | 21 (2) | 24 (3) | 34 (3) |
| | | M | 11 (2) | 21 (1) | 30 (2) | 32 (3) | 41 (2) |
| Upward rotation | Overhead reach | F | – | 9 (1) | 23 (2) | – | 40 (2) |
| | | M | – | 15 (2) | 33 (1) | – | 46 (2) |
| Upward rotation | Forward transfer | F | 2 (1) | 8 (1) | – | – | 15 (2) |
| | | M | 4 (2) | 16 (2) | – | – | 24 (2) |

beneficial for shoulder musculoskeletal disorders (*Seitz et al., 2011*) so these findings may contribute to the explanation of greater injury risk in females (*Cimas et al., 2018*; *Mcbeth, Jones & Associate, 2007*; *Wijnhoven et al., 2006*).

Previous work by our research group compared sex and age-based differences in kinematics-only in this exact WRAFT protocol (*Waslen, Friesen & Lang, 2023*). The only significant scapular angle difference in the previous study was present in scapular tilt during the Forward Transfer. There were no upward rotation differences in the previous work, unlike the current study. The discrepancies in findings between these two studies are likely due to differences in samples: *Waslen, Friesen & Lang (2023)* also tested the effects

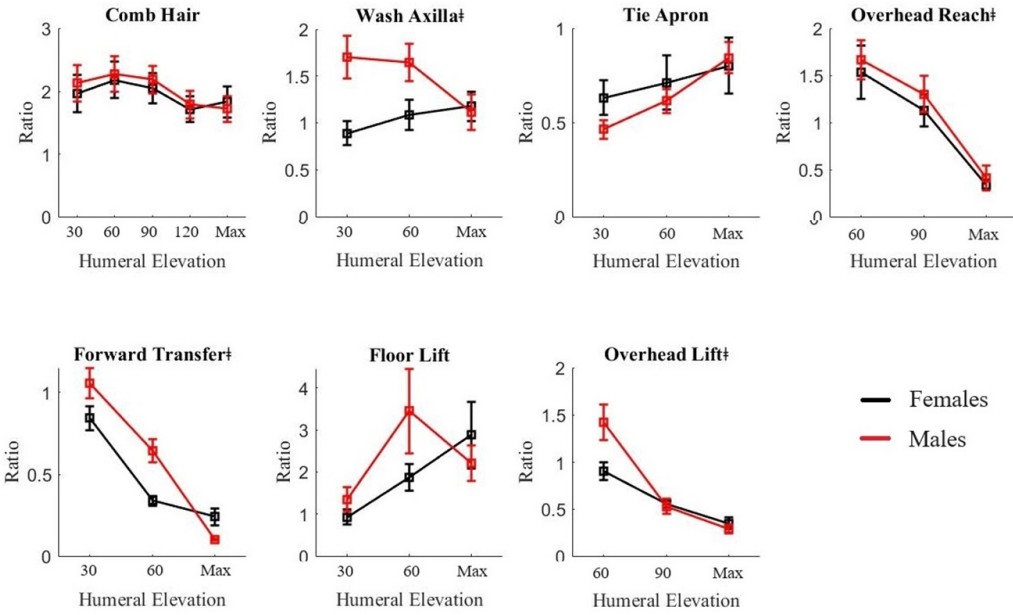

**Figure 4** **Mean +/- 95% CI MT/LT activation ratio for females (black line) and males (red line) for all functional tasks.** An asterisk (*) indicates significant interaction of sex and humeral elevation level ($p < .05$); # indicates significant main effect of sex ($p < .05$); ‡ indicates significant main effect of humeral elevation level ($p < .05$).

of age, so adults between the ages of 18 and 65 were recruited, while the current study used a convenience sample technique and all participants were under 30 years of age. Age may also cause an increase in stiffness and reduction in range of motion (*Barnes, Van Steyn & Fischer, 2001*), which may have masked movement pattern differences present in younger females and males. However, the conflicting results of the two studies do suggest upward rotation differences should be interpreted with caution when applying to the healthy population. Previous research also reports mixed results when comparing upward rotation between sexes in planar elevation (*Nagamatsu et al., 2015*; *Picco, Vidt & Dickerson, 2018*; *Schwartz et al., 2016*). Further investigation of factors that may influence the effect of sex on scapular upward rotation and associated muscle activation ratios, such as laxity, strength, or age, would help to elucidate this relationship.

Activation ratios changed with humeral elevation in both sexes. Changes in activation have been reported in previous work during planar arm elevation (*Contemori, Panichi & Biscarini, 2019*; *Hawkes et al., 2012*; *Spall, Ribeiro & Sole, 2016*). Both UT/LT and UT/SA tend to decrease with increasing humeral elevation (*Contemori, Panichi & Biscarini, 2019*). In planar elevation, both ratios started over 2.0 (UT double activity of LT/SA) and decreased to around 1.0 (equal activity) (*Contemori, Panichi & Biscarini, 2019*). However, the current study demonstrates that, while functional tasks may have a similar overall decreasing pattern, the lower trapezius may be more activated in the functional tasks as indicated by starting values lower than 2.0 at the lower humeral elevations and decreasing from there (Tie Apron the exception). Lower ratios may occur in functional tasks due to increased

co-contraction for stability and precision, as these tasks were goal-directed and often loaded (*Bohunicky et al., 2021*). Future research investigating activation patterns should consider these relative activation patterns during functional movements.

Defining scapular muscle activation ratios, and associated scapular orientations, of a control group during functional tasks provides foundational work for understanding shoulder movement during functional tasks. Muscle activation and shoulder kinematics are similar between the sexes for many tasks, but select significant differences are present, warranting the continued investigation and inclusion of sex as a factor in biomechanical analyses. These data can also direct future research with pathological groups. For example, LT and SA are not as active as the UT in lower humeral elevations even in a healthy group, as demonstrated by substantially higher ratios at lower humeral elevations (*Contemori, Panichi & Biscarini, 2019*). As previous work suggests that individuals with shoulder pain or disorders already have overactive upper trapezius (*Umehara et al., 2018*) or impaired lower trapezius (*Michener et al., 2016*), it is possible activation ratios will be more imbalanced in lower humeral elevations in a non-normative group, which could cause harmful kinematic alterations and increase injury risk during functional task performance. Future work will explore this avenue of research. Finally, the findings from this study also have applications to as normative data. The reported ratios can be used for future comparisons to non-normative groups. Just as previous analyses of scapular kinematics and muscle activity in healthy groups during planar arm elevation have served as reference points (*Ludewig, Cook & Nawoczenski, 1996*; *McClure et al., 2001*) for relevant research, these findings can provide an initial comparison for investigations of activation of both men and women in various functional movements. This is one of the first studies to reported activation ratios during this range of functional tasks, and future researchers can use this information to provide context and guide interpretation.

## Limitations

There are some limitations of this work to consider. First, as mentioned, this study was powered based on kinematic data; a larger sample may be needed to preclude Type B error for muscle activation comparison. Despite this, some statistical differences between sexes were detected, indicating the utility of this investigation to identify important sex-based differences during functional tasks. Additionally, other relevant factors that may aid in the explanation of kinematic differences, such as flexibility, were not assessed in this study. Participants were also not assessed for physical fitness level, participation in a physical training session prior to the data collection, or participation in a long-term physical training program that could effect muscle shortening or flexibity, which could influence our results. Physical fitness and training patterns should be considered in future research. Finally, normalization limitations always need to be considered with electromyography, especially when creating ratios. Only one round of MVCs were employed, to mitigate potential fatigue effects, but two or three rounds may be more effective for attaining a true maximum The serratus anterior, in particular, can be difficult to fully activate (*Ekstrom, Soderberg & Donatelli, 2005*); however, this was a young, healthy sample, which is an ideal population for MVCs, supporting this methodological choice.

## CONCLUSIONS

This study is one of the first investigations to define scapular muscle activation ratios during functional tasks. These ratios can serve as normative data for for future investigations of shoulder muscle activation during upper limb-focused functional tasks. Activation ratios were different between sexes for the Tie Apron task, which is a task that requires extension and rotation at the glenohumeral joint. Other scapular kinematic differences were also present between sexes, despite lack of other statistically significant findings for activation ratios. Sex should continue to be a factor in kinematic analyses, and further work is needed to confirm how muscle activation and scapular movement vary with sex.

### Funding

This work was supported by the University of Saskatchewan College of Medicine and the Natural Sciences and Engineering Research Council (#027-25634). The funders had no role in study design, data collection and analysis, decision to publish, or preparation of the manuscript.

### Grant Disclosures

The following grant information was disclosed by the authors:
University of Saskatchewan College of Medicine.
Natural Sciences and Engineering Research Council: 027-25634.

### Competing Interests

The authors declare there are no competing interests.

### Author Contributions

- Angelica E. Lang conceived and designed the experiments, analyzed the data, prepared figures and/or tables, authored or reviewed drafts of the article, and approved the final draft.
- Annaka Chorneyko performed the experiments, analyzed the data, prepared figures and/or tables, authored or reviewed drafts of the article, and approved the final draft.
- Vivian Heinrichs performed the experiments, authored or reviewed drafts of the article, and approved the final draft.

### Human Ethics

The following information was supplied relating to ethical approvals (i.e., approving body and any reference numbers):

The University of Saskatchewan granted Ethical approval to carry out the study within its facilities (Ethics Approval: Bio #3796).

### Data Availability

The raw data is available in the Supplemental Files.

## Supplemental Information

Supplemental information for this article can be found online at http://dx.doi.org/10.7717/peerj.17728#supplemental-information.

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
