# Peer review of "Comparing and characterizing scapular muscle activation ratios in males and females during execution of common functional movements"

_PeerJ, doi:10.7717/peerj.17728_

## Round 0.1 · original submission · Minor Revisions

Dear Authors

We would like to express our gratitude for submitting your manuscript entitled "Comparing and characterizing scapular muscle activation ratios in males and females during execution of common functional movements" to our journal.

The reviewers acknowledged the quality of your work and identified a few areas that could be enhanced to further strengthen your argument and contribution to the scientific community. Their suggestions and critiques were valuable, and thus, we kindly request that you revise your manuscript according to the reviewers' suggestions.

Should you have any questions or require further clarification regarding the revisions requested, please do not hesitate to contact us. We once again appreciate your work and collaboration in this process.

Best regards,

Alexandre Medeiros

Reviewer 1 ·

Basic reporting

no comment

Experimental design

I suggest that the authors add the importance of this study in line 76. Who could benefit from the results of this study?
I suggest the authors add the hypothesis of the study on line 81.

Methods
With regard to the study participants, the authors could provide a lot of information. I had some doubts about this. I suggest that the authors answer them and add them to the study.
1- What was the physical fitness level of the sample? Were the men and women sedentary, physically active or trained? This information is unclear and could influence the results of the study.
2- The volunteers could have performed vigorous physical exertion before the electromyography assessment session or could have been engaged in a physical training program (resistance exercise or flexibility exercises), which could increase or decrease the level of muscle shortening and influence the results. This information is not clear from the study.
3- How did the authors consider the volunteers to be healthy people? Did they use any physiological markers or questionnaires to classify them as healthy?
4- Were any exclusion criteria adopted? Were any volunteers excluded from the study?
5- I recommend that the authors add the current statistical power of the sample size to lines 87-87.

Validity of the findings

The authors have added the p-values, but I suggest adding the effect size for significant interactions.

Discussion
I suggest the authors add a paragraph reporting the usefulness/applicability of the research.

Additional comments

no comment

Annotated reviews are not available for download in order to protect the identity of reviewers who chose to remain anonymous.

Reviewer 2 ·

Basic reporting

I suggest expanding the anatomophysiological aspects that justify the difference between the sexes.

Experimental design

The manuscript it presents coherence with the scope of the PeerJ, having scientific relevance and meets a knowledge gap.
In the sample calculation, authors need to add information so that it can be reproduced. How many groups? How many measurements?.

Validity of the findings

In conclusion, I suggest expanding the practical/clinical application of the study.

Additional comments

What is the level of training of the study participants?

---

## Round 0.2 · accepted · Accept

Thank you for addressing the Reviewers issues and comments. I am pleased to recommend your amended manuscript for publication. We look forward to receiving future manuscripts from your group. Thanks, A/Prof Mike Climstein

Reviewer 1 ·

Basic reporting

No comments

Experimental design

No comments

Validity of the findings

No comments

Additional comments

I congratulate the authors on their efforts to make adjustments to the article.

Annotated reviews are not available for download in order to protect the identity of reviewers who chose to remain anonymous.

Reviewer 2 ·

Basic reporting

The authors responded adequately to all comments.

Experimental design

The authors responded adequately to all comments.

Validity of the findings

The authors responded adequately to all comments.

Additional comments

The authors responded adequately to all comments.